# Profile of Self-Reported Physical Tasks and Physical Training in Brazilian Special Operations Units: A Web-Based Cross-Sectional Study

**DOI:** 10.3390/ijerph17197135

**Published:** 2020-09-29

**Authors:** Eduardo Marins, Ossian Barbosa, Eduardo Machado, Robin Orr, Jay Dawes, Fabrício Del Vecchio

**Affiliations:** 1Superior School of Physical Education, Federal University of Pelotas, Pelotas 96055630, Brazil; fabricioboscolo@gmail.com; 2Federal Highway Police Department, Brasilia 71200029, Brazil; ossian.barbosa@prf.gov.br; 3Federal Police Department, Brasilia 70610902, Brazil; schneider.esm@dpf.gov.br; 4Tactical Research Unit, Bond University, Gold Coast, QLD 4229, Australia; rorr@bond.edu.au; 5Department of Health and Human Performance, Oklahoma State University, Stillwater, OK 74074, USA; jay.dawes@okstate.edu

**Keywords:** questionnaire, job performance, police officers

## Abstract

There is limited research examining the physical tasks that Brazilian special policemen groups can perform in the line of duty. The aims of this study were to (a) identify the occupational tasks of specialist police personnel serving in the Rapid Response Group (GRR) and Tactical Operations Command (COT), and to profile the frequency, difficulty, and importance of these tasks, and (b) to explore the current physical training these special operations police units (SOPUs) officers undertake to maintain their operational fitness. Univariate analysis for numerical variables (mean and standard deviation (SD)), as well as the absolute and relative frequencies for categorical variables were performed. Two Brazilian SOPUs performed a questionnaire with demographic, performance, and physical training sections. A total of 78 respondents (24 of the GRR and 54 of the COT) completed the survey. “Standing and/or sitting with complete equipment for long periods in different climatic conditions”, and “lifting/pushing/pulling heavier objects” were the most frequent and difficult occupational tasks of both SOPUs, respectively. “Shooting a long weapon” and “breaking a door” were the most important for GRR and COT, respectively. All officers undertook regular physical training (~9 h/week), in an unstructured manner, without supervision, and planning of physical training is carried out autonomously (COT) or a mixture of autonomously and directed (GRR), with the main objectives of developing aerobic endurance and muscle strength. It is important that SOPUs teams train their members’ physical capabilities to perform the tasks identified in this study, as well as follow a structured, supervised, and planned physical training program.

## 1. Introduction

In Brazil, crime has increased both in quantity (absolute number) and diversity (types of criminal activity) [1]. As the nature of crime changes and multiplies, public security institutions must adapt and train their workers, also known as ‘tactical professionals’ to meet these challenges [2,3,4]. Daily tasks performed in police work may range between low-intensity activities (i.e., prolonged time sitting in the police car or querying databases) to dynamic high-intensity physical tasks (i.e., a foot pursuit negotiating obstacles, such as fences, and subsequently restraining a non-compliant offender) [2,5,6]. The diversity of these tasks requires a combination of physical capabilities by police officers [7,8]. Subsequently, achieving minimum levels of physical fitness to perform essential job tasks is of interest within tactical populations.

Police officers are also required to wear personal protective equipment (PPE, such as a ballistic vest) and carry various other equipment items for police work, such as handcuffs, secondary weapons, flashlight, baton, among others [9,10]. These loads, weighing around 10–12 kg, may negatively affect the occupational performance [10,11]. Furthermore, to meet threats beyond the capability of the general police officer, many organizations have created special operations police units (SOPUs). With additional specialist equipment (e.g., weaponry, respirators, breaching equipment, ballistic shields, and battering rams), these SOPU officers are often required to carry loads exceeding 20 kg and at times over 40 kg [2,12,13].

In Brazil, police officers who work in SOPUs (e.g., the Rapid Response Group (GRR) of the Federal Highway Police (PRF) and the Tactical Operations Command (COT) of the Federal Police (PF)) are referred to as specialist police. Such specialized teams must be prepared to perform activities with different frequencies and physical requirements than those of general duties police. International studies have sought to identify and profile the occupational tasks performed by these units in the United States, New Zealand, and Australia [2,5,12]. These studies investigated the frequency, difficulty, and importance of job tasks, as well as the physical training performed by these specialist police [2,5,12]. For example, Davis et al. [2] identified that Special Weapons and Tactics (SWAT) officers may be required to perform a multitude of tasks during operations, including climbing obstacles or barriers, dragging injured team members, and lifting various objects.

To perform these tasks successfully, Davis et al. [2] indicated that muscle power and strength were the most important components for the successful performance of job tasks among SWAT officers. These results were confirmed by a similar study with police officers from an Australian special operation unit [5]. In addition to muscular strength and power, Robinson et al. [14] identified that specialist police officers required a high level of aerobic fitness to complete load carriage events. Apart from the direct relationship between officer fitness and task performance, an indirect relationship is also noted. Officers who are injured or ill, for example, would be unable to attend shifts or complete given work tasks. Moreover, fitness profiles have a wide range of applications, such as return from injury programs and to reduce the risk of reinjury [15]. Overall, these specialist police require levels of fitness greater than other police and the general population [15,16].

Given the uniqueness of these specialist tasks and fitness requirements, little is known about the physical training practices of specialist police teams in Brazil. A recent systematic review involving the characterization of the physical profile among tactical populations of elite tactical units found only three studies, out of 14 included in the review, with tactical police populations, with none from Brazil [15]. Variations in task requirements have been found to exist in a single police unit [17] as well as variations in fitness standards between police units [18,19]. As such, identifying and profiling the tasks that specialist police from both GRR and COT units perform is of importance if their physical training practices are to be based on their specific job needs.

The aims of this study were to (a) identify the occupational tasks of specialist police personnel serving in GRR and COT, and to profile the frequency, difficulty, and importance of these tasks, and (b) to explore the current physical training these SOPU officers undertake to maintain their operational fitness.

## 2. Materials and Methods

This study was reported in accordance with the Strengthening the reporting of observational studies in epidemiology (STROBE) statement [20].

### 2.1. Study Design

A cross-sectional study was conducted employing an online questionnaire sent to all police officers working in the GRR (PRF) and COT (PF) SOPUs in Brasília, during the period from March to December 2019.

### 2.2. Settings

First, two of the authors of this survey, who also worked at the PRF (OSB) and PF (ESM), received authorization from the responsible sectors of each agency (Coordination of Operations and Special Resources, PRF, and Coordination of the Command of Tactical Operations, PF) to present the study proposal to the specialist police of the SOPUs and to send them the questionnaire online via their functional emails (provided by the police chiefs). Officers received the information and were invited to participate in the research anonymously.

### 2.3. Participants

The target population of this study was made up of male police officers active in the specialist police units of GRR and COT, PRF and PF respectively, which were located in Brasília, in the Federal District, Brazil. In total, approximately 30 and 70 personnel of the PRF and PF respectively, perform their functions in the respective groups of specialist policing. There are currently no women in these units, a limitation noted in several studies in specialist personnel [12,14,21,22].

In order to be eligible to participate in this study, police officers had to be effectively employed in the role of GRR or COT specialist policing. Any personnel who had not performed, at the time of applying the questionnaire, in the functional role of specialist policing for more than three months were excluded from the survey.

### 2.4. Procedures

The data were collected through a self-administered online questionnaire, in Brazilian Portuguese, hosted on the Google platform (https://www.google.com/forms/) and distributed to all participants via their preferred and monitored e-mail. The online questionnaire consisted of three sections.

The first section included the gaining of informed and voluntary consent followed by questions to capture sociodemographic data, being: sex (male and female), age in complete years, educational level (doctorate, master’s, post-graduate, superior complete, high school), marital status (single, married/stable union, widowed), duration of police service in full years, data on the structural characteristics of the specialist police units (length of service, number of police officers, number of teams, and police officers in each team), the weight of the individual PPE carried, also known as personal kit, body mass in kg, height in cm, and the percentage of fat, if known, and its method of measurement.

The second section contained a list of tasks that were developed from previous literature [2,5,12] and in consultation with experts on the subject. Thus, the study participants were asked to rate the frequency (scale from 1 to 7, with 1 “always” and 7 “never”), the importance (scale from 1 to 7, with 1 being “essential” and 7 “dispensable”), and the difficulty (scale from 1 to 7, with 1 being “very difficult” and 7 “very easy”) of the listed tasks.

The third section involved aspects related to the structure of their fitness program (if any), and included questions about structure (structured, unstructured, mixed, mostly structured or mixed, but mostly unstructured), supervision (supervised, unsupervised, mixed, but mostly supervised or mixed, but mostly unsupervised), and the planning (command/management level, command/management and self-planned, self-planned, self-planned and by physical education professionals, physical education or training professional, or mixed way, i.e., command/management level, self-planned, and by a physical education professional) of their physical training, as well as the frequency (hours per week) and place (all at the base or equivalent locations, during work time, most of the base or equivalent locations, during work time, most off-site and off-hours, or all off-site and off-hours) of this training. In addition, police officers were asked to rate (scale 1 to 8, with 1 being “most important” and 8 “least important) the main parameters of physical fitness involved in the purpose of their physical training program (for example, aerobic endurance, power, muscular strength, muscular endurance, agility, balance, coordination, and flexibility).

The online survey was administered for a period of three months and was supported, in principle, by the senior command of the two specialist police units. Ethical approval of the research was provided by the Research Ethics Committee of the School of Physical Education, Federal University of Pelotas, Pelotas, Brazil (protocol number 19921119.0.0000.5313).

The sample size was determined by the number of police officers effectively employed in the units during the period of application of the study questionnaire.

### 2.5. Statistical Methods

Statistical analyses were performed using univariate analysis, informing measures of central tendency and dispersion for continuous numerical variables (mean and standard deviation (SD), respectively), as well as the absolute and relative frequencies for categorical variables. In order to deal with missing data, the analyses included only individuals who had all the data completed for all the variables necessary for the analyses.

The non-identifiable data were converted into a Microsoft Excel spreadsheet and later imported into the SPSS 20.0 software (IBM Corporation, Armonk, NY, USA) to perform all statistical analyses.

## 3. Results

Of the 102 police officers of SOPUs sampled, 24 operators were on vacation (all from COT), leaving an eligible sample of 78 specialist police officers. All 78 (100%) of the available SOPUs officers completed the questionnaire, which reflects 30.8% (*n* = 24) from GRR and 69.2% (*n* = 54) from COT, reflecting 100% and 69% of the total population of each respective SOPU. The demographic characteristics of police officers from both SOPUs are presented in Table 1.

All police officers from both SOPUs who answered the questionnaire were male. Most GRR (75%) and COT (79.6%) police officers are married or live with a partner. Regarding the education level of the GRR operators, 29.2% have postgraduate degrees and all others had some higher education. In COT, 7.4% (*n* = 4) have a master’s degree in some area, 16.7% (*n* = 9) have some form of postgraduate qualification, and the remainder have higher education qualifications. Also, the police officers reported that the number of teams and members that make up each team of GRR and COT are, respectively, two with 11.2 ± 0.8 officers per team, and three with 25.8 ± 3.2 officers per team. The body fat measurement methods and their respective percentages were also reported by 37 (47.4%) of participants (Table 2). In addition, officers reported carrying an estimated mean weight of individual tactical equipment used in operations of 18.2 ± 4.7 kg and 22.7 ± 6.0 kg for GRR and COT, respectively.

The classification of tasks listed in the online questionnaire, according to frequency, importance, difficulty, and mean of frequency and difficulty, are shown in Table 3.

Regarding the physical training of the SOPUs teams, all officers answered that they perform some form of physical training (*n* = 78) and spend 9.2 ± 4.7 h/week (GRR = 9.0 ± 4.7 h/week, COT = 9.3 ± 4.8 h/week) engaged in these activities. Regarding the training format, the GRR team reported their training as being mostly mixed, with the most frequent training being unstructured (*n* = 14), while the COT team reported their training as being mostly unstructured (*n* = 27). Only six officers (three from each tactical team) reported performing a structured form of physical training (Figure 1).

Regarding training supervision, the GRR team reported their supervision as being mostly mixed but mostly unsupervised (*n* = 16), while the COT team reported their training as being mostly unsupervised (*n* = 26) or mixed but mostly unsupervised (*n* = 25), and only two operators (only GRR team) reported the that their training was completely under the supervision of a coach (Figure 2).

Most COT officers (61.1%, *n* = 33) indicated that they themselves planned their own training, but in GRR, this planning was mostly done in a mixed manner at the command/management level, by the individual police officer, and by a physical education professional (*n* = 9) (Figure 3).

In addition, SOPU police officers were asked about the location where they routinely performed their physical training. The majority of officers (*n* = 56, 71.8%), from both special groups (*n* = 20 and *n* = 36, GRR and COT, respectively), reported that they performed their training at the base or equivalent facilities during work time. Finally, aerobic endurance and muscle strength were classified as the highest priority in the physical training of GRR and COT officers respectively, while balance and coordination were the lowest priority in both SOPUs (Table 4).

## 4. Discussion

The tasks of “standing and/or sitting with complete equipment for long periods in different climatic conditions” and “lifting/pushing/pulling objects (heavier than an adult man)” were classified as more frequent and more difficult respectively, in both units (GRR and COT). The tasks of “shooting a long weapon” and “breaking a door” were classified as most important for GRR and COT, respectively. In both SOPUs, physical training (~9 h/week) is mostly performed within the workplace, in an unstructured manner and without the supervision of a trained professional. Furthermore, the planning of physical training is typically carried out autonomously (COT) or with a mixture of autonomous and directed (GRR), with the main objectives of training being the development of aerobic fitness and muscular strength.

The age and sex of the police officers in this study were similar to those reported among other specialist teams [5,12,13,18]. Furthermore, the reported loads carried were greater than non-specialist police officers (~10 kg) [9,10,11] and similar to those of other specialist officers (i.e., 20 kg) [2,12]. These findings reiterate the need for strength, power, and aerobic fitness to perform essential job tasks to offset the increased physical burden created by dawning this equipment [14,23]. The importance of aerobic fitness is further supported when considering that officers who were successful in completing the selection process for a specialized police unit performed a 2.4 km loaded movement event significantly faster than those who failed [24].

When identifying the frequency of the critical tasks performed, officers from both Brazilian units reported that “standing and/or sitting with complete equipment for long periods in varying climatic conditions” was a frequently performed task. These findings are in line with previous studies with police officers in general [25,26] and among specialist police units in Australia [5] and the United States [2]. Among the operational tasks identified in this study, the most difficult, for both specialist police units, was “lifting/pushing/pulling objects (heavier than an adult man)”. This finding corroborates the findings of Davis et al. [2], who noted that SWAT team officers reported “lift”, “push”, and “pull” something that weighed greater than 150 pounds as the three most difficult tasks they are required to perform. Based on the concept of specificity, exercises such as deadlifts and carrying heavy objects should be emphasized in the training programs of these officers. Furthermore, these exercises have been found to correlate with tasks that specialist police officers may be required to perform, like general load carriage tasks and victim rescue drags [22].

As for task importance, the responses from the SOPU of GRR (“shooting a long weapon”) differed from the COT team (“breaking a door”), which conversely agreed with other literature, notably that of Davis et al. [2]. While the reasons for these trends are unclear, they may be due to functional differences between these two specialized groups. The COT team are part of the investigative police unit, while the GRR team are part of the ostensive police. Indeed, the execution of high-risk arrest warrants are intrinsically part of the COT job, while for the GRR, this kind of task is generally performed only when working in joint operations with other agencies. These findings are consistent with those reported by Irving et al. [12], who found that these were the most common tasks reported among Specialist Tactical Police of the Australian–New Zealand Counter-Terrorism Committee. The COT team reportedly carried out a large volume of high-risk arrest warrants, in which they reported moving through an urban environment, entering buildings, dwellings, and used specific method-of-entry tools (e.g., battering ram) and tactics, and tasks requiring strength and power to perform these activities. Based on these requirements, and given the importance of such tasks, it is recommended that special police officers train and maintain physical skills related to the performance of these tasks, and that physical training programs in these populations emphasize the development of muscular strength and power [2].

In this study, all respondents reported participation in some form of physical training within the workplace. This participation was typically unstructured, without the supervision of a trained professional, saw officers planning their own physical training (COT) or having their training planned in a mix fashion (i.e., autonomously and directed (GRR)), and primarily focused on the development of aerobic fitness and muscle strength. These findings are partially supported by Davis et al. [2], where the majority (49%) of SWAT members reported they design their own training program, as well as perform unsupervised training (40%). Conversely, while the study by Irving et al. [12] reported that resistance training, aerobic activity, and occupational-specific activity where the leading types of training conducted, the researchers did not identify whether the training was supervised or unsupervised nor who had designed the program. As such, the results of this study can provide at least a basic consideration of these elements of their training, pending further research.

Based on the responses of the SOPUs officers in this study regarding the most frequent, difficult, and important tasks performed, as well as their physical training regimes, any explanation for a relationship between the most tasks-related and physical training on the occupational task, while not the intent of this paper, are cautiously drawn from other research in this field. Future research should be conducted to determine a testing battery to assess the efficacy of supervised, structured, and planned training programs designed to help SOPUs members improve the performance of occupational tasks.

### Limitations

While this study makes a significant contribution to the literature in both the understanding of specialist police tasks and their training requirements, some limitations should be noted. First, the findings of this investigation are based on self-reported data by the officers being studied, which has the potential to lead to recall bias. However, this form of data capture is not atypical in this population [2,12]. Finally, because this survey was conducted among Brazilian officers, the results of the generalization of these results to special units in other countries and regions may be limited. However, there are some notable similarities to the special units of other countries and as such, this research can be used to guide research conducted in different countries to determine whether these results are based on the culture of the specific unit and geographic location.

## 5. Conclusions

Based on the findings of this study, it can be concluded that the SOPUs police officers identified similar tasks in terms of frequency and difficulty of performance, but with variations between the considerations about the most important task. In addition, all officers undertook regular physical training (~9 h/week), in an unstructured manner, without supervision, and planning of physical training is carried out autonomously (COT) or a mixture of autonomously and directed (GRR), with the main objectives of developing aerobic endurance and muscle strength.

## Figures and Tables

**Figure 1 ijerph-17-07135-f001:**
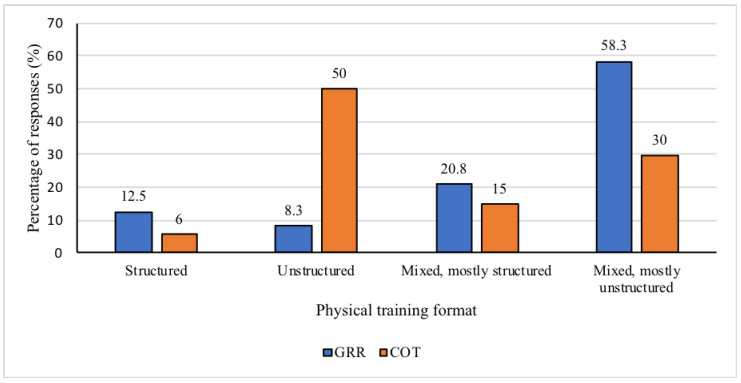
Percentage of responses on the physical training format for both special operations units, chosen from structured, unstructured, mixed (but mostly structured), or mixed (but mostly unstructured). COT = Tactical Operations Command; GRR = Rapid Response Group.

**Figure 2 ijerph-17-07135-f002:**
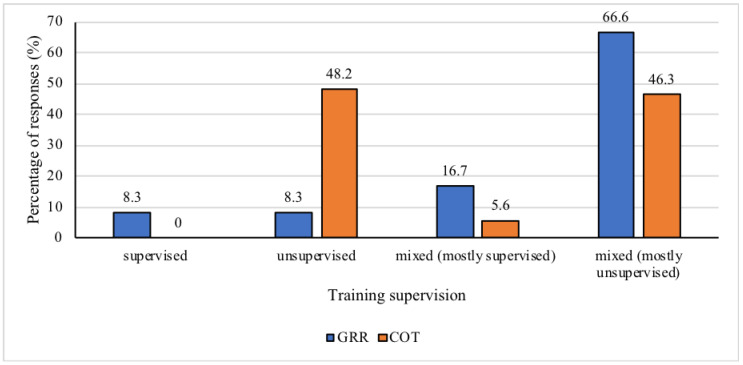
Percentage of responses to training supervision for both special operations units, chosen from supervised, unsupervised, mixed (but mostly supervised), or mixed (but mostly unsupervised). COT = Tactical Operations Command; GRR = Rapid Response Group.

**Figure 3 ijerph-17-07135-f003:**
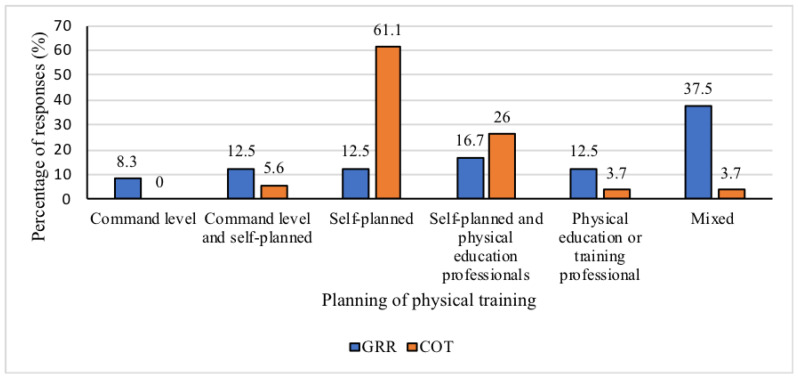
Percentage of responses on the planning of physical training of operators of special operations police units, selected from the command/management level, command/management and self-planned, self-planned, self-planned and by physical education professionals, physical education or training professional, mixed way (command/management level, self-planned, and by a physical education professional). COT = Tactical Operations Command; GRR = Rapid Response Group.

**Table 1 ijerph-17-07135-t001:** Demographic characteristics of the two study cohorts of police officers, Brasilia (Brazil), 2019.

Characteristic	GRR (*n* = 24)		COT (*n* = 54)	
	Mean ± SD	Range (min–max)	Mean ± SD	Range (min–max)
Age (years)	33.6 ± 3.5	27–41	35.5 ± 5.5	27–55
Height (cm)	175 ± 6.4	166–193	179.7 ± 6.1	168–193
Body mass (kg)	81.2 ± 8.0	69–100	84.7 ± 7.0	70–99
Police activity time (years)	6.3 ± 4.1	3–20	8.5 ± 4.7	3–23
Time as member of a SOPU (years)	2.8 ± 0.4	2–3	5.2 ± 4.8	1–20

Note: COT = Tactical Operations Command; SD = standard deviation; GRR = Rapid Response Group; max = maximum value; min = minimum value; SOPU = Special Operations Police Unit.

**Table 2 ijerph-17-07135-t002:** Reported methods for measuring body fat and respective estimated body fat percentage for both special operations units, Brasilia (Brazil), 2019 (*n* = 37).

Method	Responses	Estimated Mean % BF ± SD
Skinfold calipers	7	13.0 ± 1.7
Circumferences	2	16.0 ± 1.4
DEXA	1	16.0 ± 0
Bioelectrical impedance	27	13.0 ± 4.1

Note: BF = body fat; DEXA = dual-energy X-ray absorptiometry; SD = standard deviation.

**Table 3 ijerph-17-07135-t003:** Classification of tasks by frequency, importance, difficulty, and mean of frequency and difficulty.

Task	GRR (*n* = 24)				COT (*n* = 54)			
	Frequency	Importance	Difficulty	Mean	Frequency	Importance	Difficulty	Mean
Standing and/or sitting with all equipment for extended periods	2.3 ± 2.1	2.0 ± 1.8	4.6 ± 1.8	3.5 ± 1.4	2.1 ± 1.7	1.9 ± 1.5	4.4 ± 1.5	3.3 ± 1.1
Break a door	3.2 ± 1.4	2.1 ± 1.4	4.7 ± 1.8	3.9 ± 1.2	2.2 ± 1.7	1.6 ± 1.4	5.2 ± 1.6	3.7 ± 1.0
Walk fast or run continuously for more than 10 min	3.3 ± 1.7	1.9 ± 1.5	4.5 ± 1.7	3.9 ± 1.4	3.2 ± 1.8	2.8 ± 1.8	5.0 ± 1.8	4.1 ± 1.1
Run fast for less than 30 s	3.4 ± 2.1	2.0 ± 1.5	4.5 ± 2.1	4.0 ± 1.6	2.7 ± 1.5	2.2 ± 1.5	5.3 ± 1.8	4.0 ± 1.1
Fight/immobilize/handcuff someone who resists being arrested	4.8 ± 1.6	2.6 ± 1.8	3.3 ± 1.6	4.0 ± 1.1	3.6 ± 1.5	2.0 ± 1.6	4.4 ± 1.6	4.0 ± 1.0
Maintaining a tactical position for a long period (>15 min)	3.6 ± 2.0	2.8 ± 2.0	4.0 ± 1.6	3.8 ± 1.3	3.2 ± 1.9	2.3 ± 1.6	4.5 ± 1.6	3.9 ± 1.1
Lift/push/pull objects of average weight (approximately 20 to 70 kg)	4.1 ± 1.9	3 ± 1.8	3.4 ± 1.4	3.8 ± 1.1	2.9 ± 1.5	2.6 ± 1.6	4.7 ± 1.5	3.8 ± 1.0
Walking long distances (>1 h)	3.8 ± 1.8	2.6 ± 1.7	4.5 ± 1.7	4.1 ± 1.2	3.7 ± 1.3	2.8 ± 1.7	4.7 ± 1.6	4.2 ± 1.0
Work in small spaces/tunnels/openings/wells and hiding places	3.8 ± 1.7	3.3 ± 1.9	3.8 ± 1.5	3.8 ± 1.2	4.0 ± 1.8	3.3 ± 1.7	4.1 ± 1.5	4.1 ± 1.0
Lift/push/pull heavier objects (heavier than an adult man)	4.6 ± 1.8	3.5 ± 2.1	2.9 ± 1.7	3.8 ± 1.2	4.1 ± 1.3	3.6 ± 1.6	2.8 ± 1.5	3.5 ± 1.0
Fire a rifle	3.7 ± 1.7	1.8 ± 1.6	5.6 ± 2.0	4.6 ± 1.4	3.2 ± 1.8	1.9 ± 1.8	5.7 ± 1.7	4.5 ± 1.2
Climb/jump/cross an obstacle (such as a wall or fence) higher than you	4.1 ± 1.7	3.1 ± 1.9	3.8 ± 1.5	3.9 ± 1.2	3.6 ± 1.4	3.2 ± 1.7	4.4 ± 1.7	4.0 ± 1.1
Maintain balance to avoid falling or dodging obstacles and objects	3.7 ± 2.0	2.8 ± 1.9	4.9 ± 1.7	4.3 ± 1.2	2.8 ± 1.6	2.4 ± 1.6	5.3 ± 1.6	4.0 ± 1.1
Hanging only by the force of the arms	5.1 ± 1.6	3.9 ± 2.3	4 ± 2.1	3.8 ± 1.4	4.7 ± 1.5	3.9 ± 1.8	3.9 ± 1.7	3.6 ± 1.3
Jump/cross an obstacle (such as a wall or fence) lower than your waist height	3.7 ± 1.6	2.6 ± 1.7	5.4 ± 1.9	4.5 ± 1.2	2.8 ± 1.4	2.4 ± 1.6	5.7 ± 1.6	4.3 ± 1.0
Equip/unequip quickly	3.6 ± 1.8	3.2 ± 2.1	4.8 ± 2.0	4.2 ± 1.3	3.2 ± 1.8	2.5 ± 1.7	5.1 ± 1.5	4.2 ± 1.1
Float/traverse watercourse	5.1 ± 2.0	3.7 ± 2.3	3.4 ± 2.1	4.3 ± 1.2	4.5 ± 1.8	3.5 ± 2.0	4.9 ± 1.6	4.7 ± 1.1
Crawl	5.1 ± 2.0	4.5 ± 2.1	4.3 ± 1.7	4.7 ± 1.3	5.1 ± 1.5	4.2 ± 1.9	4.1 ± 1.9	4.6 ± 1.1
Use abseiling techniques at heights above 3 m	5.3 ± 2.2	4.5 ± 1.8	3.8 ± 1.8	4.6 ± 1.3	5.4 ± 1.5	4.6 ± 1.7	4.8 ± 1.7	5.1 ± 1.2
Hanging up using arms and legs (laziness position)	5.4 ± 1.7	4.4 ± 2.3	4.3 ± 2.0	4.8 ± 1.4	5.1 ± 1.4	4.2 ± 1.9	4.1 ± 1.8	4.6 ± 1.1

Note: COT = Tactical Operations Command; GRR = Rapid Response Group; Data are presented as mean ± standard deviation. Frequency scale range from 1 to 7, with 1 “always” and 7 “never”, importance scale ranging from 1 to 7, with 1 being “essential” and 7 “dispensable”, and difficulty scale range from 1 to 7, with 1 being “very difficult” and 7 “very easy”.

**Table 4 ijerph-17-07135-t004:** Classification of the purpose of the physical training performed by operators of Special Operations Police Units, ranging from 1 (most important) to 10 (least important).

Training Goal	GRR (*n* = 24)	COT (*n* = 54)
	Rating (Mean ± SD)	Rating (Mean ± SD)
Aerobic endurance	2.0 ± 1.7	2.9 ± 1.8
Muscle power	3.6 ± 2.0	3.4 ± 1.9
Muscle strength	3.4 ± 1.8	2.1 ± 1.6
Muscle endurance	3.3 ± 1.7	3.6 ± 1.5
Agility	4.8 ± 1.4	5.1 ± 1.5
Balance	6.6 ± 1.3	6.5 ± 1.3
Coordination	6.5 ± 1.8	6.5 ± 1.5
Flexibility	5.9 ± 1.6	6.1 ± 1.8

Note: COT = Tactical Operations Command; SD = standard deviation; GRR = Rapid Response Group.

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
