# Peer review of "Profile of Self-Reported Physical Tasks and Physical Training in Brazilian Special Operations Units: A Web-Based Cross-Sectional Study"

_ijerph, 2020, doi:10.3390/ijerph17197135_

Round 1

Reviewer 1 Report

The authors surveyed two small specialty policing units within Brazil regarding the physical tasks performed and their self reported fitness requirements/processes. 

I'm concerned that this descriptive article doesn't add anything of significance to the literature. How many officers are present in Brazil? What percentage of officers does this small group represent? Being that these officers are from highly specialized units one would assume their responses are fairly homogenous. 

Lines 164 through 167 aren't needed. Self reports of how large their unit is doesn't add anything - especially after you already listed the n for each unit. 

Table 3. This is a lot of information. How do these results compare between the two groups (beyond raw differences)? Are they significantly different between the groups (t-tests or ANOVAs could be performed)? How would these results compare to normal (non SOPU) units? 

Figure 1 & 2: Do you think these categories are truly mutually exclusive? I am concerned that there is significant overlap between the response categories. This could have been improved with perceptions questions (do the officers feel their training regimen is effective or could be improved?). 

You do a thorough job discussing the limited findings from this study and trying to extrapolate significance. While I am concerned this manuscript doesn't add much to the scientific literature, I do think this would be a better fit in a trade journal or policing magazine (e.g., PoliceOne) rather than a scholarly outlet. 

Reviewer 2 Report

The scaling presented is not intuitive with the lowest value representing the larger score.  This should be reversed.

Why present a mean of frequency, importance and difficulty?  Importance is a different construct.  A mean of frequency and difficulty may be more useful.

Some information should be presented in the background on the importance of fitness to the health and wellness of officers.  The way the article reads, the only important objective is to do the tasks, not to prevent injury or to instill the confidence to do the job.

Reviewer 3 Report

Dear all

Congratulations on your article. In the attached file there are some tips for improving your article

Regards

The reviewer

Round 2

Reviewer 1 Report

Thank you for your responses to my previous comments.  I still feel this research doesn't generalize outside of the very small, homogenous units you surveyed. Had this study expanded beyond these groups and compared them to general officers it could have had implications beyond pure description of the two units. 

Author Response

As for the generalization for, a paragraph about the limitations and generalization of the findings of this research was inserted.

Unfortunately, the purpose of this study was not to compare with general officers, but to describe the tasks performed by police officers working in special operations units. Maybe in a future study we can make these comparisons.

Thank you!